# Type 2 diabetes and age-related cognitive decline over 40 years in Danish men–A cohort study based on the Danish Aging and Cognition (DanACo) cohort

Gunhild Tidemann Okholm[1,2,3]*, Marie Grønkjær[2], Jørgen Rungby[4,5], Erik Lykke Mortensen[3], Merete Osler[2,6]

**1** National Institute of Public Health, University of Southern Denmark, Copenhagen, Denmark, **2** Centre for Clinical Research and Prevention, Copenhagen University Hospital – Bispebjerg and Frederiksberg, Copenhagen, Denmark, **3** Section of Environmental Health, Department of Public Health, University of Copenhagen, Copenhagen, Denmark, **4** Translational Type 2 Diabetes Research, Steno Diabetes Center Copenhagen, Herlev, Denmark, **5** Department of Clinical Medicine, University of Copenhagen, Copenhagen, Denmark, **6** Section of Epidemiology, Department of Public Health, University of Copenhagen, Copenhagen, Denmark

* gunh@sdu.dk

## Abstract

### Aim

The extant literature on type 2 diabetes and cognitive decline is based on short cognitive follow-ups and assessments of baseline cognitive ability after diagnosis. The objective was to investigate the influence of type 2 diabetes on cognitive decline over a period of on average 44 years.

### Materials and methods

This cohort study included 5,147 men from the Danish Aging and Cognition cohort consisting of a late mid-life (mean age 64.2 years) follow-up of men with intelligence test scores (IQ) available from statutory conscription board examinations in young adulthood (mean age 20.4 years). Follow-up included re-administration of the conscription board intelligence test and a comprehensive questionnaire. Exposure was self-reported but register-based type 2 diabetes and duration of disease were also calculated. Cognitive decline was defined as both IQ change (baseline-follow-up) and significant IQ decline based on the reliable change index (cut-off: 13.2 IQ-points). Associations were analyzed in linear and logistic regression models.

### Results

Men having type 2 diabetes had a 1.81 IQ points (95%CI:1.14,2.49) larger decline compared to men without diabetes when adjusting for baseline IQ, years of education, follow-up age, retest interval, depression, and smoking status. Moreover, type

**Data availability statement:** Data used for this study includes the DanACo cohort, which is researcher generated data, and data from national registries and is consequently stored at Statistics Denmark. Data from Statistics Denmark are not allowed to be included in publicly available de-identified databases, and consequently our project database can only be accessed at Statistics Denmark. Researchers can get access to the de-identified data through an institution approved by Statistics Denmark and with permission from the DanACo steering group. Queries regarding data access and more information about the cohort can be directed to Gunhild Tidemann Okholm [gunh@sdu.dk] or to the DanACo research team [bfh-fp-ckff-danaco@regionh.dk].

**Funding:** Support for research on the impact of type 2 diabetes and depression on cognitive decline has been granted by the Lundbeck foundation [grant number: R380-2021-1433], Helsefonden [grant number: 22-B-0196], and by the internal research funds of Frederiksberg and Bispebjerg hospitals. The funding bodies had no role in the design of the studies, nor in the collection, analysis, and interpretation of data and writing of the manuscript.

**Competing interests:** The authors have no conflicts of interest to disclose.

2 diabetes was associated with 1.42 times higher odds of a significant IQ decline and longer duration was associated with a larger, though not statistically significant, decline. The participation rate was 13.4%, and the participants were healthier and more well-educated than non-participants. To account for potential selection bias, inverse probability weights (IPW) were calculated based on baseline characteristics. The analyses applying these weights yielded similar estimates.

## Conclusion

Type 2 diabetes was associated with modestly greater cognitive decline and higher odds of a statistically significant (>13.2 IQ points) and clinically relevant decline. Finally, the alignment between main and IPW results indicates the findings are robust and likely generalizable.

## 1 Introduction

It is widely assumed that type 2 diabetes (T2D) accelerates the rate of age-related cognitive decline and increases the risk of dementia [1–4]. However, previous studies reporting more cognitive decline among individuals with T2D are characterized by methodological shortcomings including short follow-up periods, use of screening instruments to measure cognitive decline, and assessment of baseline cognitive function after diagnosis of T2D. Screening instruments, such as the Mini Mental State Examination (MMSE), only provide rough and imprecise measures of cognition and cognitive decline, and depending on the age of the study sample, perfect scores may be common and result in limited variance and skewed distributions. Post-diagnosis measures of baseline cognitive ability are problematic as they may already be affected by the disease and also because onset of T2D often occurs several years before a clinical diagnosis [5]. Moreover, several studies have shown that the baseline level of cognitive ability influences the rate of cognitive decline [6–8] and low early-life cognitive ability has been associated with a higher risk of T2D [9–11]. To date only one study has taken premorbid cognitive ability into account [12]. This study found that the association between T2D and poorer cognitive functioning in old age could be ascribed to a life-long lower cognitive ability among individuals with T2D rather than to T2D causing cognitive impairment. Thus, the lack of studies taking premorbid cognitive ability into account could suggest that differences in initial cognitive levels might partly explain the observed association between T2D and cognitive decline.

The well-established association between T2D and depression [13,14] and the association between depression and cognitive functioning [15] and dementia [16] make depression a potential explanatory factor in the association between diabetes and cognitive decline. However, the link between T2D and depression, and their combined effect on cognitive decline, is unclear. Both diseases are associated with dysfunction of the HPA-axis [17] and chronic low grade inflammation [18], which are factors that may link T2D and cognitive decline. Moreover, treatment of both disorders has been suggested to modify the risk of dementia [19,20].

The primary aim of the present study was to investigate the influence of T2D on cognitive changes over an average of 44 years using a draft board intelligence test as baseline and a re-administration of the same intelligence test in a cohort of middle-aged Danish men. Additional aims were to investigate the influence of baseline cognitive ability, measured in young adulthood, and to analyze the effect of disease duration on cognitive changes, and the potential synergistic effect of having both T2D and depression.

## 2 Materials and methods

The study was based on the Danish Aging and Cognition (DanACo) cohort comprising 5340 men who have participated in one of two cognitive follow-up studies: the *Lifestyle and Cognition follow-up study 2015* (LiKO-15) and the *Diabetes and Cognition Follow-up study 2019* (DiaKO-19) [21]. They were designed to study cognitive decline and were based on re-assessments of cognitive ability using military intelligence scores (IQ) from the statutory conscription board examination (mean age 20.4) as baseline and with a late mid-life follow-up (mean age 64.2). The DanACo cohort is thus unique in providing a direct measure of cognitive changes over an average of 44 years (p1,p99: 35.5, 53.2). The data collections for LiKO-15 and DiaKO-19 were carried out from September 2015 through June 2017 and from August 2019 through May 2022, respectively. For more details on the establishment of the DanACo cohort see Grønkjær et al [21].

Of the 5340 participants, all follow-up data for a total of 157 participants was lost due to server malfunction during two shorter time periods resulting in the deletion of data from all men participating during these periods For the present study, 18 participants with type 1 diabetes and 18 participants with missing information on either baseline IQ, education, or smoking status were excluded, leaving a total study sample of 5147 men.

### 2.1 Type 2 diabetes (T2D)

Both self-reported and register-based measures of T2D were available. Information on **self-reported T2D** was derived from the follow-up questionnaires of the LiKO-15 and DiaKO-19 studies. Both studies included questions on "physician diagnosis of diabetes" and "age of diagnosis". The LiKO-15 questionnaire asked about diabetes without specifying type. Consequently, age at diagnosis was used to categorize those diagnosed before age 35 as type 1, and those diagnosed at age 35 or later as type 2. The DiaKO-19 questionnaire included type specific questions. However, 19 participants responded yes to both type 1 and type 2 diabetes. Consequently, the same age criteria were used for DiaKO-19 as for LiKO-19. **Duration of diabetes** was calculated as the difference between self-reported age at diagnoses and age at follow-up and was categorized into three categories: 1) 0–9 years, 2) 10–19 years, and 3) 20 years or more.

For sensitivity analyses and comparisons of participants and non-participants, a **register-based measure of T2D** was derived using information on hospital diagnosis of T2D and redemption of prescriptions for antidiabetic medication from national registers (for details on the included registers, ICD-8/10 codes, and ATC codes, see Supplementary Table S1 in S1 Appendix). Duration of register-based T2D was calculated as the number of years between the date of first hospital diagnosis/prescription for antidiabetic treatment and the follow-up date and was categorized in the three categories as described above.

Within the study population, there was a high concordance between register-based and self-reported T2D (Supplementary Table S2 Table in S1 Appendix) with similar proportions across the two measures (14.8% vs. 14.2%). Only 1.1% (n=47) of the study population had a register-based diagnosis of T2D without a self-reported T2D and using the register-based measure as golden standard the sensitivity of the self-reported T2D was 93.6%.

### 2.2 Cognitive decline

Cognitive ability was assessed by the conscription board intelligence test, the Børge Prien's Prøve (BPP), administered at baseline and follow-up. The BPP comprises four subtests (letter matrices,19 items; verbal analogies, 24 items; number series,17 items; geometric figures,18 items) each to be completed within a fixed time period (a total of 45 minutes) [22].

The items of the BPP have remained unchanged since its implementation in 1957. The DanACo participants completed a paper-and-pencil version at baseline and a computerized version at follow-up. The total BPP-score (ranging from 0–78) was included from both baseline and follow-up and it has been demonstrated to correlate r = .82 with a full-scale IQ-score from the Wechsler Adult Intelligence Scale [23]. Individual subtest-scores were only available from follow-up. The BPP test scores from baseline and follow-up were linearly transformed to an IQ scale with a sample mean of 100 and a standard deviation of 15 at baseline. Two measures of cognitive decline were derived: **IQ changes** were defined as the difference between the baseline and follow-up IQ scores; a **significant decline in IQ scores** was defined based on the Reliable Change Index (RCI) – developed to estimate the amount of change in a measure needed to exceed the extent of change that could be due to measurement error alone [24]. RCI was calculated by dividing the change in score by the standard error of the difference score (estimated using the baseline standard deviation of the BPP-score of 9.5 and the estimated test reliability (omega) of 0.82, see Grønkjær et al 2024 [21]). An absolute RCI of 1.96 or more was interpreted as statistically significant and resulted in a cut-off of 13.2 IQ points signifying a significant change in IQ score.

## 2.3 Covariates

**Age at follow-up** was calculated based on date of birth and date of follow-up participation. **Retest interval length** was calculated as the difference between age at conscription board and follow-up examinations. **Baseline BMI** (kg/m$^2$) was calculated based on height and weight from the conscription board examination. **Years of education** was constructed as a combined measure of school education and post-school vocational training and education retrieved from the follow-up questionnaire. **Smoking status** was defined as ever, former, or current based on self-reported information on ever smoking and smoking during the past 12 months at follow-up. **Depression** was defined as yes/no based on a question on ever having had a "physician diagnosis of depression" in the follow-up questionnaire. For sensitivity analyses, a **register-based measure of depression** was derived based on psychiatric hospital diagnoses and redemption of antidepressants retrieved from national registers (for more details see Supplementary Table S1 in S1 Appendix).

In addition to the covariates above, **year of birth, age at invitation to follow-up,** and baseline school education were included in analyses comparing baseline characteristics of the study sample and DanACo non-participants. **Baseline school education** was categorized into 'low' (intermediate school, 10th grade or less or craft learned, industrial or trade skill), 'medium' (lower secondary school, extended graduation exam or first year of upper secondary school completed), and 'high' (second year of upper secondary school completed or more).

## 2.4 Statistical analysis

First, the study sample characteristics were presented according to T2D status and differences were tested using X$^2$ test for categorical variables and one-way ANOVA for continuous variables (Table 1). Secondly, the association between T2D and cognitive changes was investigated in linear regression models and the odds of having a significant decline was investigated in logistic regression models. Potential interactions between T2D and 1) depression, 2) baseline IQ, and 3) study origin (LiKO-15 vs. DiaKO-19) were investigated in analyses including interaction terms in fully adjusted models and using the contrast command to test whether the effect of T2D was significant irrespective of the level of the moderator. The estimates from analyses including an interaction term are presented in tables stratified on the potential moderator in Table 2, for depression, and Supplementary Tables S3 and S4 in S1 Appendix for baseline IQ and study origin, respectively. Three main regression models were analyzed (Table 3): a **Crude model** with no adjustment, **Model 1:** adjusted for baseline IQ score, follow-up age, retest interval, and years of education, and **Model 2** with additional adjustment for depression and follow-up smoking status. Thirdly, the association between duration of T2D and IQ changes/significant decline was investigated among men having a self-reported T2D (n = 761) (Table 4). The following sensitivity analyses were conducted using both linear and logistic regression: 1) additional adjustment for baseline BMI in a subsample (n = 4,825) with information

**Table 1. Characteristics of the study sample of 5,147 middle-aged Danish men by self-reported type 2 diabetes status.**

| | Total (n = 5,147) | Type 2 diabetes | | p-value |
|---|---|---|---|---|
| | | No (n = 4,386) | Yes (n = 761) | |
| Baseline IQ, Mean (SD) | 100.0 (15.0) | 100.7 (14.7) | 95.9 (16.1) | <0.001 |
| Follow-up IQ, Mean (SD) | 93.8 (14.5) | 94.7 (14.1) | 88.4 (15.7) | <0.001 |
| IQ change, Mean (SD) | −6.2 (9.7) | −6.0 (9.6) | −7.6 (9.7) | <0.001 |
| Reliable change in IQ, N (%) | | | | <0.001 |
| Yes | 1104 (21.4) | 901 (20.5) | 203 (26.7) | |
| No | 4043 (78.6) | 3485 (79.5) | 558 (73.3) | |
| Age at baseline (yrs.), Mean (SD) | 20.4 (2.1) | 20.4 (2.1) | 20.1 (1.9) | <0.001 |
| Age at follow-up (yrs), Mean (SD) | 64.4 (4.2) | 64.2 (4.2) | 65.7 (3.7) | <0.001 |
| Retest interval, Mean (SD) | 44.0 (4.3) | 43.7 (4.3) | 45.5 (4.0) | <0.001 |
| Years of education, Mean (SD) | 13.7 (2.5) | 13.9 (2.5) | 12.9 (2.6) | <0.001 |
| Smoking status, N (%) | | | | 0.081 |
| Never | 1691 (32.9) | 1467 (33.4) | 224 (29.0) | |
| Former | 2631 (51.1) | 2227 (50.8) | 404 (53.1) | |
| Current | 825 (16.0) | 692 (15.8) | 133 (17.5) | |
| Depression, N (%) | | | | 0.198 |
| Yes | 825 (16.0) | 691 (15.8) | 134 (17.6) | |
| No | 4322 (84.0) | 3695 (84.2) | 627 (82.4) | |
| Baseline BMI*, N (%) | | | | <0.001 |
| <18.5 | 340 (7.0) | 299 (7.3) | 41 (5.7) | |
| 18.5-24.9 | 4087 (84.7) | 3552 (86.5) | 535 (74.5) | |
| 25-29.9 | 365 (7.6) | 241 (5.9) | 124 (17.3) | |
| >=30 | 33 (0.7) | 15 (0.4) | 18 (2.5) | |

Abbreviations: IQ: Intelligence Quotient; SD: Standard Deviation, BMI: Body Mass Index.

*Due to missing information on either height or weight at conscription, there are only 4825 men with information on baseline BMI.

**Table 2. Adjusted\* associations between self-reported type 2 diabetes and IQ changes and having a significant decline in IQ stratified by depression status.**

| | No depression (n = 4322) | | Depression (n = 825) | |
|---|---|---|---|---|
| **IQ changes** | | | | |
| T2D | *B* | (95% CI) | *B* | (95% CI) |
| No | 0 | – | 0 | – |
| Yes | −2.16*** | (−2.94,-1.38) | −0.32 | (−1.97,1.33) |
| **Significant decline in IQ** | | | | |
| T2D | OR | (95% CI) | OR | (95% CI) |
| No | 1 | – | 1 | – |
| Yes | 1.62*** | (1.30,2.01) | 0.96 | (0.60,1.53) |

Abbreviations: IQ: Intelligence Quotient; T2D: Type 2 diabetes; *B*: unstandardized beta coefficients; CI: Confidence Interval; OR: Odds ratio.

*Adjusted for baseline IQ, age at follow-up, retest interval, years of education, smoking status, baseline BMI, and including an interaction term between Type 2 diabetes and depression.

\* $p < 0.05$, ** $p < 0.01$, *** $p < 0.001$.

**Table 3. Associations between self-reported type 2 diabetes and IQ changes and having a significant decline in IQ in Danish men (n = 5,147) in unadjusted and adjusted linear- and logistic regression analyses.**

| | Main analyses | | | | | | Sensitivity analyses | |
| --- | --- | --- | --- | --- | --- | --- | --- | --- |
| | Crude | | Model 1 | | Model 2 | | Model 3† | |
| **IQ changes** | | | | | | | | |
| T2D | *B* | (95% CI) | *B* | (95% CI) | *B* | (95% CI) | *B* | (95% CI) |
| No | 0 | | 0 | | 0 | | 0 | |
| Yes | −1.59*** | (−2.33,-0.84) | −1.83*** | (−2.51,-1.16) | −1.81*** | (−2.49,-1.14) | −1.84*** | (−2.55,-1.12) |
| **Significant decline in IQ** | | | | | | | | |
| T2D | OR | (95% CI) | OR | (95% CI) | OR | (95% CI) | OR | (95% CI) |
| No | 1 | | 1 | | 1 | | 1 | |
| Yes | 1.41*** | (1.18,1.68) | 1.44*** | (1.19,1.74) | 1.42*** | (1.18,1.72) | 1.47*** | (1.20,1.79) |

Abbreviations: IQ: Intelligence Quotient; T2D: Type 2 diabetes; *B*: unstandardized beta coefficients; CI: Confidence Interval; OR: Odds ratio.

Model 1: adjusted for baseline IQ, age at follow-up, retest interval, and years of education.

Model 2: adjusted for model 1 + smoking status, and depression.

Model 3: adjusted for model 2 + baseline BMI.

* $p < 0.05$, ** $p < 0.01$, *** $p < 0.001$.

†Due to missing information on BMI, model 3 only includes 4825 observations.

Increment in $R^2$: Model 1 = 0.44%, Model 2 = 0.43%, Model 3 = 0.42%.

**Table 4. Associations between duration of type 2 diabetes and IQ changes and having a significant decline in IQ, respectively, among men having a self-reported type 2 diabetes (n = 761) from unadjusted and adjusted linear and logistic regression analyses.**

| | n | Main analyses | | | | | | Sensitivity analyses | |
| --- | --- | --- | --- | --- | --- | --- | --- | --- | --- |
| | | Crude | | Model 1 | | Model 2 | | Model 3† | |
| **IQ changes** | | | | | | | | | |
| Duration of T2D | | *B* | (95% CI) | *B* | (95% CI) | *B* | (95% CI) | *B* | (95% CI) |
| 0–9 yrs | 345 | 0 | | 0 | | 0 | | 0 | |
| 10–19 yrs | 309 | −1.12 | (−2.61,0.38) | −0.60 | (−1.98,0.78) | −0.69 | (−2.07,0.68) | −0.83 | (−2.28,0.62) |
| >=20 yrs | 107 | −2.54* | (−4.66,-0.43) | −1.61 | (−3.57,0.35) | −1.64 | (−3.60,0.32) | −1.80 | (−3.84,0.24) |
| **Significant decline in IQ** | | | | | | | | | |
| Duration of T2D | | OR | (95% CI) | OR | (95% CI) | OR | (95% CI) | OR | (95% CI) |
| 0–9 yrs | 345 | 1 | | 1 | | 1 | | 1 | |
| 10–19 yrs | 309 | 1.20 | (0.84,1.71) | 1.06 | (0.72,1.56) | 1.07 | (0.72,1.58) | 1.09 | (0.73,1.63) |
| >=20 yrs | 107 | 1.87** | (1.17,2.98) | 1.53 | (0.91,2.57) | 1.52 | (0.90,2.57) | 1.51 | (0.88,2.58) |

Abbreviations: IQ: Intelligence Quotient; T2D: Type 2 diabetes; *B*: unstandardized beta coefficients; CI: Confidence Interval; OR: Odds ratio; yrs: years

Model 1: adjusted for baseline IQ, age at follow-up, retest interval, and years of education

Model 2: adjusted for model 1 + smoking status, and depression.

Model 3: adjusted for model 2 + baseline BMI.

* $p < 0.05$, ** $p < 0.01$, *** $p < 0.001$

†Due to missing information on BMI, model 3 only includes 686 observations.

Increment in $R^2$: Model 1 = 0.29%, Model 2 = 0.31%, Model 3 = 0.37%.

on baseline weight and height (Model 3 in Tables 3 and 4), 2) repeating the fully adjusted analyses (Model 2) of Tables 3 and 4 using the register-based measure of T2D (Table 5), and 3) repeating the same analyses with adjustment for the register-based measure of depression (Table S6 in S1 Appendix). Finally, baseline characteristics were compared between

**Table 5. Adjusted associations between register-based type 2 diabetes, duration of the disease, and IQ changes (B and 95% confidence intervals) and having a significant decline in IQ score (odds ratios and 95% confidence intervals), respectively.**

| | | IQ changes | | Significant decline in IQ | |
|---|---|---|---|---|---|
| | | Model 2 | | Model 2 | |
| | *N* | *B* | (95% CI) | OR | (95% CI) |
| **Full sample (n = 5147)** | | | | | |
| Register-based T2D | | | | | |
| No | 4402 | 0 | | 0 | |
| Yes | 745 | −2.18*** | (−2.86,-1.49) | 1.57*** | (1.30,1.90) |
| **Among men with register-based type 2 diabetes (n = 732)** | | | | | |
| Duration of register-based T2D | | | | | |
| 0–9 years | 382 | 1 | | 1 | |
| 10–19 years | 298 | 0.46 | (−0.92,1.84) | 1.00 | (0.69,1.46) |
| >=20 years | 65 | −2.97* | (−5.48,-0.47) | 2.02* | (1.07,3.81) |

Abbreviations: IQ: Intelligence Quotient; T2D: Type 2 diabetes; *B*: unstandardized beta coefficients; CI: Confidence Interval; OR: Odds Ratio.

Model 2: adjusted for baseline IQ, age at follow-up, retest interval, years of education, smoking status, and depression.

* $p < 0.05$, ** $p < 0.01$, *** $p < 0.001$.

non-participants and the study population and we attempted to account for potential selection bias by calculating inverse probability weights (IPWs), applying these weights to the data to create a pseudo population mimicking the source population, and then rerunning the main analyses. The IPWs were calculated as 1 minus the log-odds of participating based on a regression model including individual-level characteristics hypothesized to be associated with the likelihood of participating and with either the exposure or outcome. The included characteristics were IQ, education and BMI from the baseline assessment and age at time of invitation to the study. The comparison of baseline characteristics and the results based on the pseudo population are presented in Supplementary Tables S7 and S8 in S1 Appendix. Results from linear regression models were presented as unstandardized coefficients. The linear regression model assumptions were evaluated graphically, and no violations were observed. All analyses were performed using Stata 18.

## 2.5 Ethics approval

All methods were carried out in accordance with relevant guidelines and regulations. The DanACo data-collection projects were submitted for ethics approval by the Committee on Health Research Ethics in the Capital region, but the Committee ruled that according to Danish law (Scientific Ethical Committees Act (in Danish: Komiteloven), article 14, paragraph 2) approval was not required as the studies did not involve collection of biological material.

All participants received written and oral information about the background and aim of the project, that participation was voluntary, and about their right to withdraw their consent at any time during or after the follow-up examination. All participants gave their informed consent by reading and agreeing to a consent statement on the computer before the intelligence test and questionnaire were initiated.

## 3 Results

Table 1 presents the study sample characteristics by self-reported T2D status. The mean follow-up age was 64.4 years (SD = 4.2) and the mean retest interval was 44.0 years (SD = 4.3). Overall, 14.8% (n = 761) reported T2D and men with T2D had lower mean IQ scores at both baseline and follow-up and experienced a larger IQ decline (Mean IQ decline = −7.6, SD = 9.7) compared to men without T2D (Mean IQ decline = −6.0, SD = 9.6). Men with T2D had a higher proportion with a significant decline in IQ (26.7% vs. 20.5%), they had fewer years of education, and a smaller proportion with

a BMI < 18.5 and markedly higher proportions with a BMI of 25–29.9 or of 30 or higher, compared to men without T2D. The distribution across categories of smoking was similar across T2D status with about a third being never smokers (29.0% vs. 33.4%) and almost half being former smokers (53.1% vs. 50.8%). The proportion of men with previous depression was slightly higher among men with T2D (17.6% vs. 15.8%), but the difference was not statistically significant.

Table 2 presents the results from the interaction analyses exploring the potential synergistic effect between T2D and depression. The interaction term between T2D and depression was only marginally significant in analyses of both IQ changes (p = 0.046) and significant decline in IQ (p = 0.044). T2D was associated with a statistically significant larger decline and higher odds of a significant decline among men with no self-reported depression, but the association was not statistically significant for the smaller group of men with no self-reported depression. Given that the interaction terms were only marginally significant, we have decided to include depression as a confounder in the adjusted models. There was no indication of differential effects of T2D across levels of baseline IQ (Table S3 in S1 Appendix) or study origin (Table S4 in S1 Appendix).

Table 3 presents the results from the linear and logistic regression analyses. The linear regression analyses revealed a statistically significant larger mean IQ decline among men with compared to without T2D in both crude and adjusted analyses, and around 0.4% of the variance in IQ changes was attributable to T2D status. The association was strengthened after adjustment for potential confounders, particularly those of model 1, and post hoc analyses revealed that the association became stronger with adjustment for baseline IQ and education but was attenuated with adjustment for age at follow-up and retest interval (Supplementary Table S5 in S1 Appendix). In the fully adjusted model (Model 2), the estimated mean IQ decline was −7.74 (95%CI: −8.37,-7.12) for men with T2D and −5.93 (95%CI: −6.19,-5.68) for men without T2D. T2D was also associated with 1.4 times higher odds of having a significant decline in IQ compared to not having T2D and the association remained statistically significant with adjustment for covariates. Overall, the sensitivity analyses revealed similar associations when adjusting for baseline BMI (Model 3) and the register-based measure of depression (Supplementary Table S6 in S1 Appendix). Analyses using the register-based measure of T2D revealed slightly stronger associations (Table 5).

Finally, among the 761 men having T2D, there was a clear tendency towards larger decline in IQ and higher odds of having a significant decline with longer duration (Table 4). The results were not statistically significant for self-reported T2D, but for men with register-based T2D there was a statistically significant larger decline and the odds of having a significant decline was doubled for men with a duration of 20 years or more compared to a duration of 0–9 years (Table 5).

Overall, the study population had more favorable characteristics at the conscription board examination including higher IQ, higher educational levels, taller height, and lower BMI compared to non-participants (Supplementary Table S7 in S1 Appendix). Non-participants had a slightly larger proportion with register-based T2D compared to the study population (16.9% vs. 14.2%). For both non-participants and the study population, those who later received a T2D diagnosis were slightly younger, had lower IQ, were slightly shorter, and had a lower proportion with high educational level and a markedly higher proportion with a BMI above 25 or of 30 or higher. After applying the inverse probability weights to account for potential selection bias, the estimated associations between self-reported T2D and IQ changes were strengthened (difference: 2.33 IQ points) and the amount of variance explained by T2D alone increased to 0.65%. The association between T2D and having a significant decline in IQ was attenuated but remained statistically significant with 1.34 times higher odds among men with T2D (Table S8 in S1 Appendix).

## 4. Discussion

### 4.1 Main findings

In this cohort of 5147 men with cognitive ability scores from the same intelligence test administered in young adulthood and on average 44 years later, we found that having T2D was associated with a larger decline in cognitive ability (difference: 1.81 IQ points) compared to not having T2D when adjusting for baseline cognitive ability, age at follow-up, retest

interval, years of education, depression, and follow-up smoking status. Moreover, there was a clear tendency of a larger decline with longer diabetes duration, and in the analysis using the register-based measure of T2D a doubling of the odds of having a significant decline was found for men with a duration of more than 20 years compared to men with a duration of 0–9 years. The association was robust and stable in sensitivity analyses with further adjustment for baseline BMI, register-based depression, and when using a register-based measure of T2D. The difference was modest and smaller than the commonly used cut-off of half a standard deviation (7.5 IQ points) as indicator of a clinically relevant difference [25,26]. However, an almost 1.4 times higher odds of having a significant decline (≥13.2 IQ points) was found among men with T2D compared to those without. Evaluation of the generalizability revealed that the study population had more favorable baseline characteristics on cognitive ability, education and BMI than non-participants, but the distribution of baseline characteristics according to register-based T2D status was similar for both populations. Furthermore, similar associations were found when attempting to account for selection bias by applying inverse probability weights calculated based on characteristics known for both participants and non-participants, namely age at time of invitation to follow-up and baseline IQ, education and BMI.

## 4.2  Comparison with previous studies

Although T2D has consistently been associated with poorer cognitive functioning in midlife and old age, the findings on the association between T2D and cognitive decline are mixed [27]. One review reported a clear association between T2D and cognitive decline [1]. This review from 2005 identified 25 prospective observational studies on the association between T2D and cognitive decline or dementia. Of the 25 studies, 21 reported on changes in cognitive scores over time and the authors found a clear trend of greater decline in individuals with T2D compared to those without. This aligns with the results of our study even though the present study had considerably longer follow-up and included a pre-morbid baseline measure of cognitive ability. Moreover, the review identified 10 studies that used a cut-off for cognitive decline (reduction of 6.6% to 11.5% or using the 10th-20th percentile). The authors presented pooled estimates for analyses using cut-off scores for six studies using the Mini Mental State Examination (MMSE) and two studies using the Digit Symbol Substitution (DSS) test. They found a pooled odds ratio (OR) of 1.2 (95% CI: 1.05,1.4) for cognitive decline based on MMSE and an OR of 1.7 (95% CI: 1.3,2.3) based on DSS. This aligns with the OR of 1.4 found in our study using the Reliable Change Index to identify individuals with a significant decline. In contrast to the present study, the studies in the review by Cukierman and colleagues included participants who were 45 years or older at baseline, with T2D assessed at the same time as the baseline cognitive assessment, and with shorter follow-up ranging between 2 and 7 years.

A more recent review from 2020 [27] identified 17 studies focusing on the association between T2D (17 of the 17 studies) or pre-diabetes (4 of the 17 studies) with cognitive changes in individuals aged 65 years or older at intake. Only three of the studies were also included in the review from 2005. In this review, the association between T2D and cognitive changes was less clear. Eleven out of the 17 studies found a statistically significant association with larger decline among individuals with T2D. The authors did not report any pooled estimates. The review focused on individuals aged 65 or older at intake and the mean follow-up time was again shorter compared to the present study (range: 2–15 years). However, the authors found that follow-up length and prevalence of T2D did not affect associations.

## 4.3  Interpretation of findings

Apart from acute diabetes-related changes [28], the process by which T2D induces cognitive dysfunction is thought to be slow and progressive and it is known that T2D onset may occur several years prior to its clinical diagnosis [5]. In contrast to most prior studies, the present study included a baseline measure of cognitive ability assessed in young adulthood, making it highly unlikely that the baseline cognitive ability, was affected by T2D. Consistent with the only prior study including a pre-morbid cognitive measure [12], we found that men with T2D had lower baseline IQ than those without T2D, indicating that lower cognitive ability may increase T2D risk. Unlike the prior study, however, the IQ difference widened at follow-up, and

adjusting for baseline IQ strengthened the association, suggesting that T2D may accelerate cognitive decline and that the difference in late-life cognitive ability of men with and without T2D is not just a result of a lifelong lower ability among men with T2D. Despite the long follow-up, the difference between men with and without T2D was modest (1.81 IQ points), but this might reflect the relatively young age (mean age: 64.4 years) at follow-up. While a difference of 1.81 IQ points lacks clinical relevance and may not be noticeable for the individual, the difference in cognitive decline between men with and without T2D may increase with increasing age which is supported by the larger relative risk for dementia found with increasing age [29]. Moreover, the odds of having a significant decline in IQ was substantially higher among men with T2D compared to men with no T2D.

Our a priori hypothesis of a synergistic effect of having both T2D and having had depression was not supported in the analysis. The reasons are not clear, but the all-male population may blunt the importance of depression. Co-morbid depression is more common in women with T2D [14], and our interaction analysis may not have had statistical power to evaluate effect modification by depression. Moreover, there could be a problem with misclassification, as unrecognized depression may be more common among men due to cultural norms and gender-related differences in symptom recognition and help-seeking behavior [30]. Finally, early-life low cognitive ability has been associated with a higher risk of incident T2D, with the main suggested mechanism being poorer health-related behaviors among individuals with low cognitive ability [9]. It has thus been suggested that the association between T2D and cognitive decline could, at least partly, be ascribed to differences in the rate of cognitive decline depending on the initial level of cognitive ability. However, we found no interaction between T2D and baseline IQ, suggesting that the association between T2D and cognitive decline was not dependent on the initial level of cognitive ability. On the other hand, we did find baseline IQ to be a potential confounder suggested by the strengthening of the association after adjustment (Supplementary Table S5 in S1 Appendix).

## 4.4  Strengths and weaknesses

The present study has several strengths. The DanACo cohort was designed to study predictors of cognitive decline, has a relatively large sample size, and with cognitive decline assessed using the same comprehensive intelligence test administered twice with several decades between baseline and follow-up. Compared to previous studies, a major strength is the availability of a baseline cognitive ability measure in young adulthood ensuring minimal influence of T2D or prediabetes on the baseline score. Relying on self-reported information on physician diagnosis of T2D might have led to potential misclassification of exposure. However, there was a high level of concordance between the self-reported information and the information on T2D derived from nationwide registers on hospital diagnoses and prescribed medications (insulin and oral antidiabetic drugs). Using the register-based measure as golden standard, the sensitivity of the questionnaire-based measure of T2D was 93.6%. Undiagnosed T2D is another likely source of misclassification of exposure and under the assumption of misclassification at random this would mean that our estimates are underestimated. The proportion having T2D was much higher in the DiaKO-19 study (around 20%) compared to the LiKO-15 study (around 8%) because the DiaKO-19 study was designed to study T2D which was clearly stated in the invitation letter. However, in our interaction analyses, we found no indication of differences in the association between T2D and cognitive decline between the two studies (Table S4 in S1 Appendix). The main weakness of the DanACo cohort is the low participation rate (14.3%), increasing the risk of selection bias. As was shown, the study population had more favorable characteristics at the conscription board examination compared with non-participants and DanACo participants have previously been shown to have a lower Charlson Comorbidity Index score (indicating a lower degree of somatic co-morbidity) and fewer psychiatric hospital contacts compared to non-participants [21]. This suggests that the study population represents a group of men that is healthier and more educated than the background population. To attempt to account for the potential selection bias, we calculated the probability of participating based on the age at time of invitation to follow-up and baseline IQ, education and BMI of both participants and non-participants and created a pseudo population by applying inverse probability weights (1-log odds of participating) to the study population. This way individuals with characteristics that were underrepresented received a higher IPW and thus a greater weight in the analyses and vice versa with

individuals having characteristics that were overrepresented in the study population. The comparable results obtained after applying inverse probability weights suggest that the impact of selection bias is minimal. This strengthens the validity of our conclusions and implies that the observed associations may reflect those in the underlying source population. Another limitation of the study was the loss of follow-up data for 157 participants due to server malfunction during two shorter time periods. The participants chose their own examination times and were not invited in any specific order, so the server malfunctions were not related to participant characteristics and could thus not result in selection bias. Furthermore, 322 men had missing information on baseline BMI caused by some conscription board districts not registering weight at all. We thus consider the missing observations to be missing at random wherefore the estimates of the sensitivity analyses may be underestimated but not biased. We had no information on physical inactivity and diet, which are both associated with T2D [31–33] and modest predictors of cognitive decline [34–36], and thus could have explained part of the association. While the BPP-test items remained unchanged, the format was changed from paper-and-pencil at baseline to a computerized format at follow-up. In a study of all men examined by the conscription board from 2006 through 2019, it has been shown that the change in format coincided with a 1.5-point drop in mean IQ, suggesting that the format shift reduced scores slightly [37]. However, we have no reason to suspect that the format change affected men with and without T2D differently, and therefore the change is unlikely to have introduced bias. Finally, the DanACo cohort is an all-male cohort. Findings are mixed when it comes to potential sex-differences in cognitive decline [8,38–40], however, the two reviews of the association between T2D and cognitive decline [1,27] did not report any sex-differences suggesting that the modestly larger decline associated with T2D in the present study may potentially be generalizable to women. However, as co-morbid depression is more common in women with T2D it might have been possible to establish a clear interaction between T2D and depression among women.

## 5 Conclusions

In this unique cognitive aging cohort including more than 5000 men who were administered the same intelligence test with an average follow-up time of 44 years, we found significantly larger cognitive decline for men with T2D compared to men with no T2D. The difference was modest, but significant, and was comparable with findings of previous studies despite the study population being relatively young with an average age of 64 years at follow-up. Furthermore, men with T2D had substantially higher odds of having a significant decline compared to men without T2D. Finally, applying inverse probability weights to account for selection bias yielded estimates similar to those from the unweighted analyses. This consistency is reassuring and suggests that our findings are likely robust and may be generalizable to the source population.

## Supporting information

**S1 Appendix. Supplementary Tables S1-S8.**
(DOCX)

## Acknowledgments

The authors would like to thank the Danish Defense for the permission to use the military intelligence test in the follow-up examinations and the personnel at the Military Recruitment and Career – Selection and Assessment Unit for excellent collaboration during the data collections. Moreover, the authors would like to thank the project workers, including project coordinators and data collectors, for their invaluable work in conducting the data collections for the DanACo cohorts. Finally, the authors thank all men who offered their time and participated in the follow-up examinations.

## Author contributions

**Conceptualization:** Gunhild Tidemann Okholm, Marie Grønkjær, Jørgen Rungby, Erik Lykke Mortensen, Merete Osler.
**Data curation:** Gunhild Tidemann Okholm, Marie Grønkjær.

**Formal analysis:** Gunhild Tidemann Okholm.

**Funding acquisition:** Gunhild Tidemann Okholm.

**Methodology:** Gunhild Tidemann Okholm, Marie Grønkjær, Jørgen Rungby, Erik Lykke Mortensen, Merete Osler.

**Supervision:** Erik Lykke Mortensen, Merete Osler.

**Writing – original draft:** Gunhild Tidemann Okholm.

**Writing – review & editing:** Gunhild Tidemann Okholm, Marie Grønkjær, Jørgen Rungby, Erik Lykke Mortensen, Merete Osler.

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
