## [Decision Letter · Decision Letter 0]

15 Jul 2025

Dear Dr. Okholm,

Thank you for submitting your manuscript to PLOS ONE. After careful consideration, we feel that it has merit but does not fully meet PLOS ONE’s publication criteria as it currently stands. Therefore, we invite you to submit a revised version of the manuscript that addresses the points raised during the review process.

We look forward to receiving your revised manuscript.

Kind regards,

Aleksandra Klisic

Academic Editor

PLOS ONE

**Journal Requirements:**

1. When submitting your revision, we need you to address these additional requirements. Please ensure that your manuscript meets PLOS ONE's style requirements, including those for file naming. The PLOS ONE style templates can be found at https://journals.plos.org/plosone/s/file?id=wjVg/PLOSOne_formatting_sample_main_body.pdf and https://journals.plos.org/plosone/s/file?id=ba62/PLOSOne_formatting_sample_title_authors_affiliations.pdf 2. We note that you have indicated that there are restrictions to data sharing for this study. For studies involving human research participant data or other sensitive data, we encourage authors to share de-identified or anonymized data. However, when data cannot be publicly shared for ethical reasons, we allow authors to make their data sets available upon request. For information on unacceptable data access restrictions, please see http://journals.plos.org/plosone/s/data-availability#loc-unacceptable-data-access-restrictions.  Before we proceed with your manuscript, please address the following prompts: a) If there are ethical or legal restrictions on sharing a de-identified data set, please explain them in detail (e.g., data contain potentially identifying or sensitive patient information, data are owned by a third-party organization, etc.) and who has imposed them (e.g., a Research Ethics Committee or Institutional Review Board, etc.). Please also provide contact information for a data access committee, ethics committee, or other institutional body to which data requests may be sent. b) If there are no restrictions, please upload the minimal anonymized data set necessary to replicate your study findings to a stable, public repository and provide us with the relevant URLs, DOIs, or accession numbers. Please see http://www.bmj.com/content/340/bmj.c181.long for guidelines on how to de-identify and prepare clinical data for publication. For a list of recommended repositories, please see https://journals.plos.org/plosone/s/recommended-repositories. You also have the option of uploading the data as Supporting Information files, but we would recommend depositing data directly to a data repository if possible. Please update your Data Availability statement in the submission form accordingly. 3. We note that you have included the phrase “data not shown” in your manuscript. Unfortunately, this does not meet our data sharing requirements. PLOS does not permit references to inaccessible data. We require that authors provide all relevant data within the paper, Supporting Information files, or in an acceptable, public repository. Please add a citation to support this phrase or upload the data that corresponds with these findings to a stable repository (such as Figshare or Dryad) and provide and URLs, DOIs, or accession numbers that may be used to access these data. Or, if the data are not a core part of the research being presented in your study, we ask that you remove the phrase that refers to these data. 4. Your ethics statement should only appear in the Methods section of your manuscript. If your ethics statement is written in any section besides the Methods, please move it to the Methods section and delete it from any other section. Please ensure that your ethics statement is included in your manuscript, as the ethics statement entered into the online submission form will not be published alongside your manuscript.

Reviewers' comments:

Reviewer's Responses to Questions

**Comments to the Author**

1. Is the manuscript technically sound, and do the data support the conclusions?

Reviewer #1: Yes

Reviewer #2: Yes

2. Has the statistical analysis been performed appropriately and rigorously?

Reviewer #1: Yes

Reviewer #2: Yes

3. Have the authors made all data underlying the findings in their manuscript fully available?

Reviewer #1: Yes

Reviewer #2: Yes

4. Is the manuscript presented in an intelligible fashion and written in standard English?

Reviewer #1: Yes

Reviewer #2: Yes

**Reviewer #1: ** Dear Dr. Dr. Aleksandra Klisic,

Thank you for the opportunity to review the manuscript titled “Type 2 diabetes and age-related cognitive decline over 40 years in Danish men – A cohort study based on the Danish Aging and Cognition (DanACo) cohort,” submitted to PLOS ONE.

This manuscript presents a well-designed and methodologically robust investigation into the long-term association between type 2 diabetes (T2D) and cognitive decline over a 40-year period, utilizing data from the Danish Aging and Cognition cohort. A notable strength of the study is the inclusion of pre-morbid cognitive ability assessed in young adulthood, which enhances the interpretability of the longitudinal findings and provides a key methodological advantage over many prior studies in the literature.

The authors report a modest yet statistically significant difference in IQ decline among men diagnosed with T2D compared to non-diabetic counterparts, alongside an increased risk of substantial cognitive decline as measured by the Reliable Change Index. The study’s methodological rigor is further supported by the use of both self-reported and registry-based diabetes diagnoses, as well as comprehensive adjustment for a range of relevant confounding variables, including depression, educational attainment, smoking status, and body mass index (BMI). These methodological strengths contribute to the overall robustness and credibility of the findings.

I find the manuscript to be a valuable and timely contribution to the expanding body of research on cognitive aging and metabolic

health. The study is methodologically sound and conceptually relevant. However, I recommend a few minor revisions aimed at enhancing the clarity of presentation and strengthening the contextual framing of the findings within the broader literature:

1. Clarify Clinical Significance: The observed IQ difference (~1.8 points) is statistically significant but may be perceived as small. While the authors note this in the discussion, a brief elaboration on the potential functional impact—or lack thereof—would help readers interpret the practical meaning of the result.

2. Depression Interaction: The rationale for excluding depression as a moderating factor could be more clearly stated. A brief explanation in the discussion about why a synergistic effect between T2D and depression was not observed (e.g., cohort characteristics) would be welcome.

3. Participation Bias: Given the relatively low participation rate (~14%), this limitation should be emphasized more clearly in both the abstract and conclusion sections to ensure appropriate interpretation of generalizability.

4. Terminological Precision: The manuscript occasionally shifts between “IQ” and “cognitive ability.” Standardizing or clarifying the usage of these terms may improve precision for interdisciplinary readers.

5. Formatting and Flow: Some minor editorial polishing would improve clarity—especially in integrating Table 4 more directly into the narrative and improving typographic consistency.

I recommend that the manuscript be accepted pending minor revisions, as outlined above. This is a thoughtfully designed and well-contextualized study that offers a meaningful contribution to our understanding of the long-term cognitive consequences associated with type 2 diabetes.

Thank you again for the opportunity to review this work.

Sincerely,

**Reviewer #2: ** Thank you for the opportunity to review this manuscript, which investigates the association between type 2 diabetes (T2D) and cognitive decline over 40 years in Danish men using the unique Danish Aging and Cognition (DanACo) cohort. The study leverages longitudinal cognitive data from young adulthood to late mid-life, aiming to address gaps in prior literature regarding pre-morbid cognitive assessment and long-term follow-up.

Title (Page 1, Lines 2–4)

The title specifies "Danish men" but does not clarify the male-only cohort in the introduction.

Abstract (Page 1, Lines 25–46; Page 9, Lines 24–46)

The abstract states the objective as investigating T2D’s influence on cognitive decline "over a period of more than 40 years" (Lines 26–28). However, the mean retest interval is 44 years (Page 15, Line 193), and the range (35.5–53.2 years, Page 11, Line 91) suggests significant variability.

Results misalignment: The abstract reports a 1.81-IQ-point decline (Lines 38–39) but Table 2 (Page 18) cites 1.83 points.

Outcome ambiguity: "Significant IQ decline" is defined via the Reliable Change Index (RCI) (Lines 35–36), but the abstract omits the clinical relevance of the 13.2-IQ-point cutoff.

Introduction

Background/Rationale (Page 10, Lines 49–82)

The introduction hypothesizes synergistic effects of T2D and depression (Lines 81–82) but later dismisses this (Page 15, Lines 173–178) without reconciling the initial rationale.

The manuscript positions itself as unique in assessing pre-morbid cognition (Lines 62–68) but downplays the single prior study (Reference 12) addressing this (Lines 63–66).

Objectives (Page 11, Lines 77–82)

The aim to analyze "potential synergistic effect" of T2D and depression (Line 82) is not evaluated in primary results (Table 2) or discussed robustly (Page 23, Lines 337–340).

Methods

Study Design & Participants (Page 11–12, Lines 83–98)

157 participants lost due to "technical errors" (Page 11, Line 95) are excluded without detailing the nature of errors or potential bias.

The 14.3% participation rate (Page 24, Line 368) and healthier profile of participants (Page 24, Lines 369–373) are noted but not quantitatively evaluated for impact on effect estimates.

Variables (Page 12–14, Lines 99–157)

Self-reported T2D classification uses inconsistent criteria between LiKO-15 and DiaKO-19 (Page 12, Lines 103–107). This heterogeneity is not addressed in sensitivity analyses.

The BPP test switched from paper-and-pencil to computerized format between baseline and follow-up (Page 13, Line 130). Equivalence of these formats is asserted but not empirically validated.

The RCI formula uses a baseline SD of 9.5 and reliability (omega) of 0.82 (Page 13, Lines 140–142), but the source of these values is unreferenced.

Physical activity, diet, and cardiovascular comorbidities—established confounders in T2D-cognition pathways—are not adjusted for.

Statistical Methods (Page 14–15, Lines 164–190)

The marginally significant T2D-depression interaction (p=0.046, Page 15, Line 177) is dismissed without sensitivity analysis.

Post hoc analyses reveal confounding patterns (Supplementary Table S3, Page 17, Lines 216–218), but the final model (Model 2) does not include retest interval, a key covariate.

Results

Participants & Descriptive Data (Page 15–16, Lines 192–208; Table 1)

Baseline BMI categories (Page 16, Lines 198–199) include "<18.5" (underweight), but this group’s relevance to T2D is unclear. No rationale for cutoff choices is provided.

Baseline BMI has 322 missing values (Page 16, Line 207), but analyses including BMI (Model 3) do not discuss implications of reduced sample size.

Outcome Data & Main Results (Page 17–20; Tables 2–4)

The 1.81-IQ-point decline (Table 2) is termed "modest" (Page 20, Line 277), but the 26.7% vs. 20.5% rate of "significant decline" (Page 16, Line 197) is clinically meaningful yet underemphasized.

For T2D duration ≥20 years (n=107, Page 19, Table 3), nonsignificant trends (p>0.05) are reported despite point estimates suggesting clinically relevant decline (e.g., OR=1.52 for significant decline).

Results for depression as an effect modifier (Page 15, Lines 170–178) are not quantified in tables.

Discussion

Key Findings & Interpretation (Page 20–23, Lines 275–348)

Mechanisms linking T2D to cognitive decline (Page 22, Lines 316–320) are discussed without direct evidence from the study.

Generalizability to women (Page 25, Lines 383–384) is claimed despite male-only data and acknowledged sex differences in T2D-depression comorbidity (Page 23, Line 339).

Limitations & Generalizability (Page 24–25, Lines 367–384)

The assertion that baseline characteristics were "similar" between participants and non-participants with T2D (Page 20, Lines 378–380) is unsupported by data (Supplementary Table S4 not shown).

The healthier, more educated cohort (Page 24, Lines 373–374) may attenuate observed effects, but the magnitude of this bias is not estimated.

**Do you want your identity to be public for this peer review?** For information about this choice, including consent withdrawal, please see our Privacy Policy

Reviewer #1: No

Reviewer #2: No

---

## [Author Response · Author response to Decision Letter 1]

8 Dec 2025

Response to reviewers

PONE-D-25-26526

Type 2 diabetes and age-related cognitive decline over 40 years in Danish men - A cohort study based on the Danish Aging and Cognition (DanACo) cohort

I want to sincerely thank both reviewers for taking the time to review the paper. I have responded to all comments below. Page and line numbers refer to the placement in the manuscript with track changes.

Reviewers' comments:

Reviewer #1: Dear Dr. Dr. Aleksandra Klisic,

Thank you for the opportunity to review the manuscript titled “Type 2 diabetes and age-related cognitive decline over 40 years in Danish men – A cohort study based on the Danish Aging and Cognition (DanACo) cohort,” submitted to PLOS ONE.

This manuscript presents a well-designed and methodologically robust investigation into the long-term association between type 2 diabetes (T2D) and cognitive decline over a 40-year period, utilizing data from the Danish Aging and Cognition cohort. A notable strength of the study is the inclusion of pre-morbid cognitive ability assessed in young adulthood, which enhances the interpretability of the longitudinal findings and provides a key methodological advantage over many prior studies in the literature.

The authors report a modest yet statistically significant difference in IQ decline among men diagnosed with T2D compared to non-diabetic counterparts, alongside an increased risk of substantial cognitive decline as measured by the Reliable Change Index. The study’s methodological rigor is further supported by the use of both self-reported and registry-based diabetes diagnoses, as well as comprehensive adjustment for a range of relevant confounding variables, including depression, educational attainment, smoking status, and body mass index (BMI). These methodological strengths contribute to the overall robustness and credibility of the findings.

I find the manuscript to be a valuable and timely contribution to the expanding body of research on cognitive aging and metabolic

health. The study is methodologically sound and conceptually relevant. However, I recommend a few minor revisions aimed at enhancing the clarity of presentation and strengthening the contextual framing of the findings within the broader literature:

Response: Thank you for taking the time to carefully read and review our manuscript.

1. Clarify Clinical Significance: The observed IQ difference (~1.8 points) is statistically significant but may be perceived as small. While the authors note this in the discussion, a brief elaboration on the potential functional impact—or lack thereof—would help readers interpret the practical meaning of the result.

Response: We have added a sentence stating the lack of clinical relevance of a 1.81 IQ point difference: p. 20, lines 428-432.

While a difference of 1.81 IQ points lacks clinical relevance and may not be noticeable for the individual, the difference in cognitive decline between men with and without T2D may increase with increasing age which is supported by the larger relative risk for dementia found with increasing age (29).

2. Depression Interaction: The rationale for excluding depression as a moderating factor could be more clearly stated. A brief explanation in the discussion about why a synergistic effect between T2D and depression was not observed (e.g., cohort characteristics) would be welcome.

Response: We agree that the exploration of the hypothesized interaction between depression and T2D has not been presented sufficiently. We have thus decided to present the results from the stratified analyses in a new Table 2 and describe the results at the beginning of the results section. We have also elaborated the discussion of why we do not find a synergistic effect in the discussion section. P. 20 lines 434-441:

Our a priori hypothesis of a synergistic effect of having both T2D and having had depression was not supported in the analysis. The reasons are not clear, but the all-male population may blunt the importance of depression. Co-morbid depression is more common in women with T2D (14), and our interaction analysis may not have had statistical power to evaluate effect modification by depression. Moreover, there could be a problem with misclassification, as unrecognized depression may be more common among men due to cultural norms and gender-related differences in symptom recognition and help-seeking behavior(30).

3. Participation Bias: Given the relatively low participation rate (~14%), this limitation should be emphasized more clearly in both the abstract and conclusion sections to ensure appropriate interpretation of generalizability.

Response: We have stated the low participation rate in both abstract and conclusion. Furthermore,

following the suggestion of reviewer 2, we have attempted to account for potential selection bias by

applying inverse probability weights based on the predicted probability of participating in the

follow-up using baseline characteristics (IQ, education, and BMI) and age at invitation to the

follow-up examination. The analyses of Table 3 (originally tables 2 and 3) were rerun applying

these weights (Supplementary Table S8) and the estimated associations were comparable, thus

strengthening the validity or our conclusions and implying that the observed associations may

reflect those in the underlying source population.

4. Terminological Precision: The manuscript occasionally shifts between “IQ” and “cognitive ability.” Standardizing or clarifying the usage of these terms may improve precision for interdisciplinary readers.

Response: Thank you for making us aware of these shifts in terminology. We have adjusted the manuscript so that we use cognitive ability when we refer to the theoretical concept of intelligence, primarily in the Introduction and Discussion, and IQ when referring to the actual scores. We introduce IQ in the Methods section under the description of the two operationalization’s of cognitive decline and IQ is primarily used in the results section, but also when referring to specific estimates in the Abstract and Discussion.

5. Formatting and Flow: Some minor editorial polishing would improve clarity—especially in integrating Table 4 more directly into the narrative and improving typographic consistency.

Response: We have carefully edited the manuscript and the layout of the tables has been streamlined and unnecessary elements have been moved to the footnotes, to make the tables more easily readable.

I recommend that the manuscript be accepted pending minor revisions, as outlined above. This is a thoughtfully designed and well-contextualized study that offers a meaningful contribution to our understanding of the long-term cognitive consequences associated with type 2 diabetes.

Reviewer #2: Thank you for the opportunity to review this manuscript, which investigates the association between type 2 diabetes (T2D) and cognitive decline over 40 years in Danish men using the unique Danish Aging and Cognition (DanACo) cohort. The study leverages longitudinal cognitive data from young adulthood to late mid-life, aiming to address gaps in prior literature regarding pre-morbid cognitive assessment and long-term follow-up.

Response: Thank you for taking the time to carefully read and review our manuscript.

Title (Page 1, Lines 2–4)

The title specifies "Danish men" but does not clarify the male-only cohort in the introduction.

Response: It has now been specified in the aim that the study is based on a cohort of middle-aged Danish men. P. 5 lines 86-87:

The primary aim of the present study was to investigate the influence of T2D on cognitive changes over an average of 44 years using a draft board intelligence test as baseline and a re-administration of the same intelligence test in a cohort of middle-aged Daish men.

Abstract (Page 1, Lines 25–46; Page 9, Lines 24–46)

The abstract states the objective as investigating T2D’s influence on cognitive decline "over a period of more than 40 years" (Lines 26–28). However, the mean retest interval is 44 years (Page 15, Line 193), and the range (35.5–53.2 years, Page 11, Line 91) suggests significant variability.

Response: To emphasize the variability in retest interval length, we have rephrased it so that it now states: “over a period of on average 44 years”, and we have done so throughout the abstract and main text. However, we have decided to keep the term “over 40 years” in the title to keep the title short.

Results misalignment: The abstract reports a 1.81-IQ-point decline (Lines 38–39) but Table 2 (Page 18) cites 1.83 points.

Response: It has now been corrected so that all references to the main results of Table 3 (originally Table 2) throughout the abstract and main text is now from model 2 (1.81 IQ points).

Outcome ambiguity: "Significant IQ decline" is defined via the Reliable Change Index (RCI) (Lines 35–36), but the abstract omits the clinical relevance of the 13.2-IQ-point cutoff.

Response: We have specified the cut-off (cut-off: 13.2 IQ-points) in the abstract and have rephrased the conclusion to put more emphasis on the clinical relevance of the higher odds of having a significant IQ decline. P. 2 lines 47-48.

Type 2 diabetes was associated with modestly greater cognitive decline and higher odds of a statistically significant (>13.2 IQ points) and clinically relevant decline.

Introduction

Background/Rationale (Page 10, Lines 49–82)

The introduction hypothesizes synergistic effects of T2D and depression (Lines 81–82) but later dismisses this (Page 15, Lines 173–178) without reconciling the initial rationale.

Response: We agree that the exploration of the hypothesized interaction between depression and T2D has not been presented sufficiently. We have thus decided to present the results from the interaction analyses in a new Table 2 and describe the results at the beginning of the results section. We have also elaborated the discussion of why we do not find a synergistic effect in the discussion section. P. 20 lines 434-441:

Our a priori hypothesis of a synergistic effect of having both T2D and having had depression was not supported in the analysis. The reasons are not clear, but the all-male population may blunt the importance of depression. Co-morbid depression is more common in women with T2D (14), and our interaction analysis may not have had the statistical power to detect an interaction between T2D and depression. Moreover, there could be a problem with misclassification, as unrecognized depression may be more common among men due to cultural norms and gender-related differences in symptom recognition and help-seeking behavior(30).

The manuscript positions itself as unique in assessing pre-morbid cognition (Lines 62–68) but downplays the single prior study (Reference 12) addressing this (Lines 63–66).

Response: We have elaborated on the differences of the findings of our study and the study by Mottus et al in the Discussion, p. 19 lines 419-425:

Consistent with the only prior study including a pre-morbid cognitive measure(12), we found that men with T2D had lower baseline IQ than those without T2D, indicating that lower cognitive ability may increase T2D risk. Unlike the prior study, however, the IQ difference widened at follow-up, and adjusting for baseline IQ strengthened the association, suggesting that T2D may accelerate cognitive decline and that the difference in late-life cognitive ability of men with and without T2D is not just a result of a lifelong lower ability among men with T2D.

Objectives (Page 11, Lines 77–82)

The aim to analyze "potential synergistic effect" of T2D and depression (Line 82) is not evaluated in primary results (Table 2) or discussed robustly (Page 23, Lines 337–340).

Response: We agree that the results of the analyses exploring the interaction between depression and T2D have not been presented sufficiently. We have thus decided to present the results in a new Table 2 and describe the results in the beginning of the results section.

Methods

Study Design & Participants (Page 11–12, Lines 83–98)

157 participants lost due to "technical errors" (Page 11, Line 95) are excluded without detailing the nature of errors or potential bias.

Response: The data was lost due to a server malfunction at two separate occasions during the data collection. It was the military that handled the collection and extraction of data, and the server malfunctions resulted in all data collected during the two specific time periods (each covering around 2 weeks) were not saved to the server and could thus not be recovered. The timing of the errors were random and the participants chose the time of their examination themselves, and we did not invite the participants in any particular order based on their characteristics. We have added some more details in the methods section and added a sentence in the discussion about potential bias.

Of the 5340 participants, all follow-up data for a total of 157 participants was lost due to server malfunction during two shorter time periods resulting in the deletion of data from all men participating during these periods. Page 5 lines 103-105:

Another limitation of the study was the loss of follow-up data for 157 participants due to server malfunctions during two shorter time periods. Since the participants chose their own examination times and were not invited in any specific order, the server malfunctions were not related to participant characteristics and could thus not result in selection bias. Page 22 lines 491-495.

The 14.3% participation rate (Page 24, Line 368) and healthier profile of participants (Page 24, Lines 369–373) are noted but not quantitatively evaluated for impact on effect estimates.

Variables (Page 12–14, Lines 99–157)

Response: Thank you for this suggestion. We have now attempted to account for potential selection bias by applying inverse probability weights based on the predicted probability of participating in the follow-up using baseline characteristics (IQ, education, and BMI) and age at invitation to the follow-up examination. The analyses of Table 3 (originally tables 2 and 3) were rerun applying these weights and the estimated associations were comparable, thus strengthening the validity or our conclusions and implying that the observed associations may reflect those in the underlying source population.

Self-reported T2D classification uses inconsistent criteria between LiKO-15 and DiaKO-19 (Page 12, Lines 103–107). This heterogeneity is not addressed in sensitivity analyses.

Response: It is correct that the self-reported information on T2D was not consistent in LiKO-15 and DiaKO-19. However, we did use the same criteria for distinguishing between T1D and T2D for both cohorts (T1D if age of onset was before age 35 otherwise T2D). We also did an interaction analyses including an interaction term between T2D and study origin. There was no indication of interaction. The results of this analysis were not presented in the original submission, so we have now added them to the supplementary material (see

Supplementary Table S4).

The BPP test switched from paper-and-pencil to computerized format between baseline and follow-up (Page 13, Line 130). Equivalence of these formats is asserted but not empirically validated.

Response: The change from paper-and-pencil to the computerized format was implemented in the autumn of 2010, and in a paper from 2021, I have together with colleagues, showed that the change coincided with a drop in mean IQ of 1.15 points. The mean was relatively stable from 2006 to 2010, dropped 1.5 IQ points and then remained relatively stable from 2011 to 2019. As the items have remained unchanged it suggests that the change of format is responsible for the drop in the mean number of items answered correctly. However, we have no reason to suspect that the change in format has introduced bias as we must expect that the change affects men with and without T2D equally. We have included a sentence on this matter in the Discussion, p. 23 lines 501-507:

While the BPP-test items remained unchanged, the format was changed from paper-and-pencil at baseline to a computerized format at follow-up. In a study of all men examined by the conscription board

---

## [Decision Letter · Decision Letter 1]

23 Dec 2025

Type 2 diabetes and age-related cognitive decline over 40 years in Danish men - A cohort study based on the Danish Aging and Cognition (DanACo) cohort

PONE-D-25-26526R1

Dear Dr. Okholm,

We’re pleased to inform you that your manuscript has been judged scientifically suitable for publication and will be formally accepted for publication once it meets all outstanding technical requirements.

Kind regards,

Aleksandra Klisic

Academic Editor

PLOS One

Additional Editor Comments (optional):

Reviewers' comments:

Reviewer's Responses to Questions

**Comments to the Author**

Reviewer #1: All comments have been addressed

Reviewer #3: (No Response)

2. Is the manuscript technically sound, and do the data support the conclusions?

Reviewer #1: Yes

Reviewer #3: Yes

3. Has the statistical analysis been performed appropriately and rigorously?

Reviewer #1: Yes

Reviewer #3: Yes

4. Have the authors made all data underlying the findings in their manuscript fully available?

Reviewer #1: Yes

Reviewer #3: Yes

5. Is the manuscript presented in an intelligible fashion and written in standard English?

Reviewer #1: Yes

Reviewer #3: Yes

Reviewer #1: Thank you for the thorough and constructive revision. The manuscript has improved substantially, particularly regarding (i) clearer framing of clinical relevance, (ii) explicit presentation of the T2D×depression analyses (now in Table 2), and (iii) the additional inverse-probability weighting analysis addressing potential selection bias. Overall, I believe the manuscript is suitable for publication pending minor editorial clarifications.

I have only a few remaining minor points:

1. IPW description (please clarify the weighting formula and implementation): In the revised text, the IPW computation is described in a way that is not fully standard/transparent (e.g., wording around “1 minus the log-odds …”). Please state explicitly: the participation model used, the exact weight formula, and whether any trimming/stabilization was applied. This will improve reproducibility and reader confidence.

2. Table 2 title/text consistency: Please check the labeling and wording around Table 2. The table content reflects analyses stratified by depression status, but the table title/caption appears to reference study origin. Also, one sentence in the Results appears to contain a minor wording inconsistency regarding the “smaller group …” (please verify and correct).

3. Abstract phrasing on generalizability: Given the male-only cohort and low participation rate, please keep language on generalizability appropriately cautious (even with the IPW robustness check).

With these minor edits, I would support acceptance.

Sincerely.

Reviewer #3: The revised manuscript entitled “Type 2 diabetes and age-related cognitive decline over 40 years in Danish men” has addressed the reviewers’ prior concerns in a thorough, transparent, and methodologically sound manner.

however, I have observed some minor areas for improvement.

Minor editorial issues remain, including small typographical errors and occasional inconsistencies in table headings and labeling, which would benefit from a final careful proofread.

Description of inverse probability weighting could be clarified using more standard methodological phrasing to improve transparency for readers.

Causal language in a few parts of the discussion should be further moderated to consistently reflect the observational nature of the study.

Generalizability beyond men could be framed more cautiously, emphasizing the need for sex-specific confirmation in future studies.

**Do you want your identity to be public for this peer review?** For information about this choice, including consent withdrawal, please see our Privacy Policy

Reviewer #1: No

Reviewer #3: **Yes: ** Uzair Abbas

---

## [Editor Report · Acceptance letter]

PONE-D-25-26526R1

PLOS One

Dear Dr. Okholm,

I'm pleased to inform you that your manuscript has been deemed suitable for publication in PLOS One. Congratulations! Your manuscript is now being handed over to our production team.

Kind regards,

on behalf of

Dr. Aleksandra Klisic

Academic Editor

PLOS One